# Cell sources of inflammatory mediators present in bone marrow areas inside the meniscus

Francisco Airton Castro Rocha[1]*, Virgínia Claudia Carneiro Girão[2], Rodolfo de Melo Nunes[1], Ana Carolina Matias Dinelly Pinto[1], Bruno Vidal[3], João Eurico Fonseca[3,4]

1 Departmento de Morfologia, Faculdade de Medicina, Universidade Federal do Ceará, Fortaleza, Ceará, Brasil, 2 Department of Internal Medicine, Faculty of Medicine, Federal University of Ceará, Fortaleza, Ceará, Brazil, 3 Serviço de Reumatologia e Doenças Ósseas Metabólicas, Hospital de Santa Maria, CHULN, Lisboa, Portugal, 4 Unidade de Investigação em Reumatologia, Instituto de Medicina Molecular, Faculdade de Medicina, Universidade de Lisboa, Centro Académico de Medicina de Lisboa, Portugal

* arocha@ufc.br

**Data Availability Statement:** All relevant data are within the paper and its Supporting Information files

**Funding:** The study was funded by: UID/BIM/ 50005/2019, project funded by Fundação para a

## Abstract

### Purpose

To demonstrate the production of inflammatory mediators by cells located in bone marrow spaces inside rodent menisci.

### Methods

Mice subjected to transection of the medial collateral and anterior cruciate ligaments and meniscotomy (osteoarthritis model) or to a sham procedure, as well as non-operated (naive) mice and rats, had knee joints excised. Tissues were stained with hematoxylin-eosin and tartrate-resistant acid phosphatase (TRAP). CD68$^+$ cells, inducible nitric oxide synthase (iNOS), interleukin (IL)-1β, and tumor necrosis factor (TNF) expression were detected using immunohistochemistry.

### Results

Lamellar ossified areas, bone-entrapped osteocytes and bone marrow spaces were found inside menisci of one week up to 6 months-old naïve mice, regardless of gender. Menisci from naive rats also showed the same pattern with bone marrow areas. CD68$^+$ cells were identified in bone marrow areas inside the meniscus of mice. TRAP$^+$ osteoclasts, and hematogenous precursors expressing IL-1β, TNF, and iNOS were identified inside bone marrow areas in meniscal samples from both naïve and sham operated mice. Quantitative immunoexpression of IL-1 β, TNF and iNOS was more intense, P = 0.0194, 0.0293, 0.0124, respectively, in mouse knees from mice sacrificed 49 days after being subjected to an osteoarthritis (OA) model as compared to sham operated animals.

Ciência e a Tecnologia (FCT)/ Ministério da Ciência, Tecnologia e Ensino Superior (MCTES) through Fundos do Orçamento de Estado.

**Competing interests:** The authors have declared that no competing interests exist.

## Conclusion

We provide novel data showing that rodent menisci display bone marrow areas with cells able to produce inflammatory mediators. Immunoexpression of inflammatory mediators in those bone marrow areas is significantly more pronounced in mice subjected to experimental OA

## Introduction

The menisci are semilunar shaped structures located inside the knee joint that play a relevant role in joint biomechanics. Resection of a morphologically damaged meniscus, which is assumed to cause joint instability, is associated with the development of knee osteoarthritis (OA) [1]. An injured though apparently intact meniscus is also being increasingly recognized as involved in OA development. Data from the Framingham study have shown that patients with damaged meniscus at nuclear magnetic resonance (NMR) imaging have increased prevalence of hand OA [2]. Moreover, edema in the adjacent bone marrow of the femur and tibia and lesions of the meniscus were the only joint damage parameters positively associated with knee OA development, as opposed to cartilage alterations, as reported using data from the Osteoarthritis Initiative cohort [3]. Thus, from a previously innocent bystander, the meniscus is being increasingly recognized as a structure directly involved in OA pathogenesis [4].

Two major distinct cell types compose the menisci. Fibrochondrocytes are the most abundant and predominate in the middle and inner parts of the meniscus, whereas fibroblasts are the major cells in the outer parts [2,5]. While the former are oval/round shaped cells with abundant surrounding extracellular matrix the later are immersed in a dense connective tissue. Blood and nerve cells are also found in the peripheral areas [6]. It has been shown that patients with inflammatory arthropathies present CD68[+] macrophages and lymphocytes inside the meniscus border [7]. Further, cells extracted from menisci of either normal or osteoarthritic joints were shown to have increased expression of inflammatory cytokines and metalloproteinases following stimulation with fibronectin or cytokines [8]. It was also previously demonstrated that in vitro exposure of menisci to compressive strain increased inducible nitric oxide synthase (iNOS) and interleukin (IL)-1β gene expression and nitrate production, suggesting a linkage between meniscal compressive load and inflammation inside joints [9]. However, precise identification of which meniscal cells are responsible for the production of inflammatory mediators remains to be demonstrated.

A recent report showed that bone marrow areas are found inside ossified portions of mice menisci [10]. During the observation of knee samples from mice subjected to experimental OA following meniscotomy, we found defined bone marrow cavities in some surgically lesioned menisci (our unpublished data). Given the inflammatory potential of bone marrow derived cells, we decided to perform an exhaustive evaluation of menisci obtained from mice and rat knees. Our data revealed active osteoclasts and CD68[+] macrophages inside bone marrow areas entrapped in the meniscus. Cells located in those bone marrow spaces express inflammatory mediators, which is more pronounced in an OA model in mice.

## Materials and methods

### Animals

Swiss mice (25–40 g) and Wistar rats (120 – 180g) were provided by the central animal house of the Federal University of Ceará, Fortaleza–CE, Brazil. Animals were housed in cages (6 /

cage) in temperature-controlled rooms with a 12h light/dark cycle with free access to water and food. There were 6 animals of either sex per each age group of mice or rats. One, 2, 4, and 6 weeks-old as well as 2, 4, 6, and 11 months-old non operated (naïve) mice, had their knee joints excised after euthanasia that followed anaesthesia with i.m. ketamine (50 mg/kg) and xylazine (10 mg/kg). In order to evaluate another species, knee joints from groups of 2, 4, 6, and 11 months-old rats were also collected. The joints were processed for histopathology (see below).

Osteoarthritis model–A group of six 2.5 months-old Swiss female mice was subjected to an experimental OA model, as described previously [11]. Briefly, after anaesthesia with i.m. ketamine (50 mg/kg) and xylazine (10 mg/kg) and local asepsia, the right knee joint was opened through a parapatellar medial incision with a scalpel. The patella was laterally displaced in order to expose the joint. Both the medial collateral and anterior cruciate ligaments were carefully transected in order to ease the access for transection of the medial meniscus without damaging the articular cartilage. As a control, another group of six female mice was subjected to a sham procedure, including skin incision, patella displacement, and exposure of the joint without damage to ligaments or menisci. The joint capsule and skin of mice from both groups were sutured with Vycril (6–0) and mononylon (4–0) threads, respectively. Both the sham and operated groups of mice were evaluated every other day following the surgical procedure, during routine cleaning of the cages, checking for possible behavior alterations. Body weight, food and water consumption were checked weekly until the end of the protocol. Euthanasia of the animals was processed under terminal anesthesia (i.m. ketamine/xylazine overdose). Those subjected to the surgical protocol were euthanized also under terminal anesthesia (i.m. ketamine/xylazine overdose) 21 or 49 days after the surgical procedure and had their knee joints evaluated under histopathology. All efforts were made to minimize animal suffering and the number of animals used. The protocol was approved by our local ethics committee (Comitê de Ética em Experimental Animal–Faculty of Medicine–Federal University of Ceará) that follows the rules of the Brazilian Committee on Animal Experimentation (COBEA).

## Histopathology

After fixation in 10% v / v formaldehyde solution and decalcification (5% v / v formic acid in 10% v / v formaldehyde solution), the whole joint, comprising the distal femoral and proximal tibial extremities, was processed for paraffin-embedding and hematoxylin-eosin (HE) staining. The material was serially sectioned at 5 μm in the sagittal plane of the articular surface, from the outer to the inner limits of the condyles. At every tenth section (50 μm apart), one section was removed for staining, with ten different sections for each sample. Blind (VCCG) analysis was specifically done for menisci, looking for the presence of ossified matrix and bone marrow areas in serially sectioned specimens. High-power views of the positive specimens were further analyzed to demonstrate the presence of specific bone cells.

## Tartrate-resistant acid phosphatase (TRAP) staining

TRAP staining was performed using the avidin–biotin–peroxidase method in the paraffin-embedded tissue sections mounted on poly-L-lysine-coated microscope slides. The sections were deparaffinized and rehydrated through xylene and graded alcohols. After antigen retrieval, endogenous peroxidase was blocked (30 minutes) with 3% (v/v) hydrogen peroxide and washed in phosphate-buffered saline (PBS). Sections were incubated overnight (4ºC) with primary polyclonal rabbit anti-TRAP (1:200; Santa Cruz biotechnology). All the antibodies were diluted in PBS plus bovine serum albumin (BSA). The slides were then incubated with biotinylated goat anti-rabbit diluted 1:200 in PBS–BSA. After washing, the slides were

incubated with avidin–biotin–horseradish peroxidase conjugate for 30 min, according to the protocol of the manufacturer. TRAP was visualized with the chromogen 3,3 diaminobenzidine (DAB) after 2 minutes of incubation. Negative control sections were processed simultaneously as described above but with the first antibody being replaced by 5% PBS–BSA. Slides were counterstained with hematoxylin, dehydrated in a graded alcohol series, cleared in xylene, and coverslipped. Revealing reagent system was purchased from DAKO, São Paulo, Brazil.

### Immunohistochemistry for iNOS, tumor necrosis factor (TNF), Interleukin (IL)-1β and CD68 expression

Four μm sections were prepared from paraffin-embedded knee joint tissues, as previously described. After deparaffinization, antigenic recuperation was performed with citrate buffer (pH 6.0) for 20 min. Endogenous peroxidase was blocked with 3% $H_2O_2$ for 10 min to reduce non-specific binding. The sections were incubated with anti-IL-1β (1:100), anti-TNF (1:100 Abcam) or anti-iNOS (1:200) antibodies (Abcam, São Paulo, Brazil) and diluted in DAKO antibody diluent for 1 h. Sections were then incubated for 30 min with polymer K4061 (DAKO, São Paulo, Brazil). The antibody binding sites were visualized by the incubation with DAB (DAKO, São Paulo, Brazil) solution. Semi-quantitation was done using Photo-shop, version 3.0 software (Adobe Systems; Mountain View, CA). Data from at least 3 independent experiments are expressed in number of pixels. Samples incubated without the primary antibody were used as a negative control.

Additionally, an anti-CD68 (0.5 ug/ml) (Abcam, UK) was used as a primary antibody with DAKO antibody diluent for 1h. Further, sections were incubated for 30 min with the polymer EnVision+ (DAKO, Glostrup, Denmark). Color was developed in a solution containing diaminobenzadine-tetrahydrochloride (Sigma, Missouri, USA), 0.5% $H_2O_2$ in phosphate-buffered saline buffer (pH 7.6). Slides were counterstained with hematoxylin and mounted. Slides were scanned and images acquired by a Hamamatsu NanoZoomerSQ slide scanner. Sections incubated with antibody diluent, without the primary antibody, were considered negative controls.

### Statistical analysis

Results are shown as the number and percentage of animals in each age group that present ossified matrix with structured bone marrow inside the meniscus with six animals (twelve joints) in each age group. Immunohistochemistry data are reported as means ± SEM of at least 3 independent experiments and were compared using Student's "t" test. P<0.05 was considered statistically significant.

Statistical analysis of the data was performed using Statistics for Windows, Version 16.

## Results

### Evaluation of bone marrow structures inside the meniscus

The present data describe the prevalence of ossified areas with bone marrow spaces inside menisci obtained from the knee joints of mice and rats of various age groups. Table 1 displays the percentage of menisci with bone marrow spaces inside ossified areas in mice and rats ranging from 1 week to 11 months-old. A tendency to an increase in the percentage of samples with ossified matrix as animals thrive can be observed, with 6 weeks-old mice displaying the highest prevalence of samples with bone marrow areas. Thereafter, there was a decrease in the presence of ossified matrix in older age groups with no demonstrable ossified areas in the 11 months-old mouse group.

**Table 1. Frequency of ossified matrix with bone marrow areas inside rodent menisci.**

| | Weeks | | | | Months | | | |
|---|---|---|---|---|---|---|---|---|
| | 1 | 2 | 4 | 6 | 2 | 4 | 6 | 11 |
| Mice | 1 (8.3) | 1 (8.3) | 2 (16.6) | 5 (41.6) | 5 (41.6) | 3 (29) | 2 (16.6) | 0 |
| Rat | | | | | 5 (41.6) | 6 (50) | 6 (50) | 11 (91.6) |

A similar percentage of menisci with ossified matrix and bone marrow structures was observed in 2, 4, or 6 months-old rats (Table 1). Contrary to what we noticed in mice, almost all (90%) 11 months-old rats displayed defined ossified matrix with bone marrow structures inside their menisci.

Naïve mice and rats (6 / group) were euthanized and had the knee joints excised for histology, using H&E staining. Data represent n (%) of joints with ossified matrix and bone marrow inside the meniscus across various ages (1–6 weeks; 2–11 months).

Fig 1 is representative of HE-stained meniscal samples from naive mice and rats. There is a clear demonstration at low-power view of a lamellar bone formation inside the meniscus in both species, with bone marrow cavities. A clearly defined megakaryocyte and hematogenous precursor cells in mouse and rat meniscal samples are shown in Fig 1A and 1B, respectively. Additionally, bone-resorbing, active osteoclasts, which are TRAP-stained could be found (Fig 1C and 1D).

Fig 2 illustrates CD68+ expression in cells located in the central portion of a mouse meniscus. Positive CD68 staining cells can also be seen in the highly vascularized peripheral red zone of the meniscus, as well as in the adjacent subchondral bone areas.

## Immunohistochemistry expression of IL-1β, TNF, and iNOS

Figs 3 and 4 illustrate the immunostaining for IL-1β, TNF, and iNOS in meniscal cells of naive and OA samples, respectively. Fibroblasts and fibrochondrocytes located predominantly in the red zone of naive menisci as well as cells inside bone marrow spaces stained positive for those mediators, whereas cells in the peripheral, poorly vascularized areas did not show any immunostaining (Fig 3A–3C). Similar to naive animals, samples from the sham-operated animals also showed positive immunoexpression of the inflammatory mediators in cells closer to the red zone (Fig 4A–4C). In menisci from animals subjected to experimental OA a more intense and diffuse immunostaining for IL-1β and TNF as well as for iNOS was found, including cells inside bone marrow areas (Fig 4D–4I). Those animals had a significantly higher expression of the quantified inflammatory mediators 21 and 49 days after surgery when compared to sham-operated animals (Fig 4J–4L).

## Discussion

We demonstrated that menisci from naive mice and rats of various age groups display ossified areas with bone marrow spaces. Previous studies have shown that the meniscus present ossified areas [12] but there were no attempts to estimate the prevalence of such structures neither the presence of bone marrow spaces was ever reported. Actually, in a very recent report, authors claim that those ossified areas were present exclusively in the anterior horn [10]. Our data reveal that ossified areas are relatively common in rodents and are also found within the middle part of the meniscus in adult animals. We emphasize that our observations should not be mistakenly confused with the occurrence of ossicles inside joints, present as loose bodies, which are not entrapped in the meniscus [13].

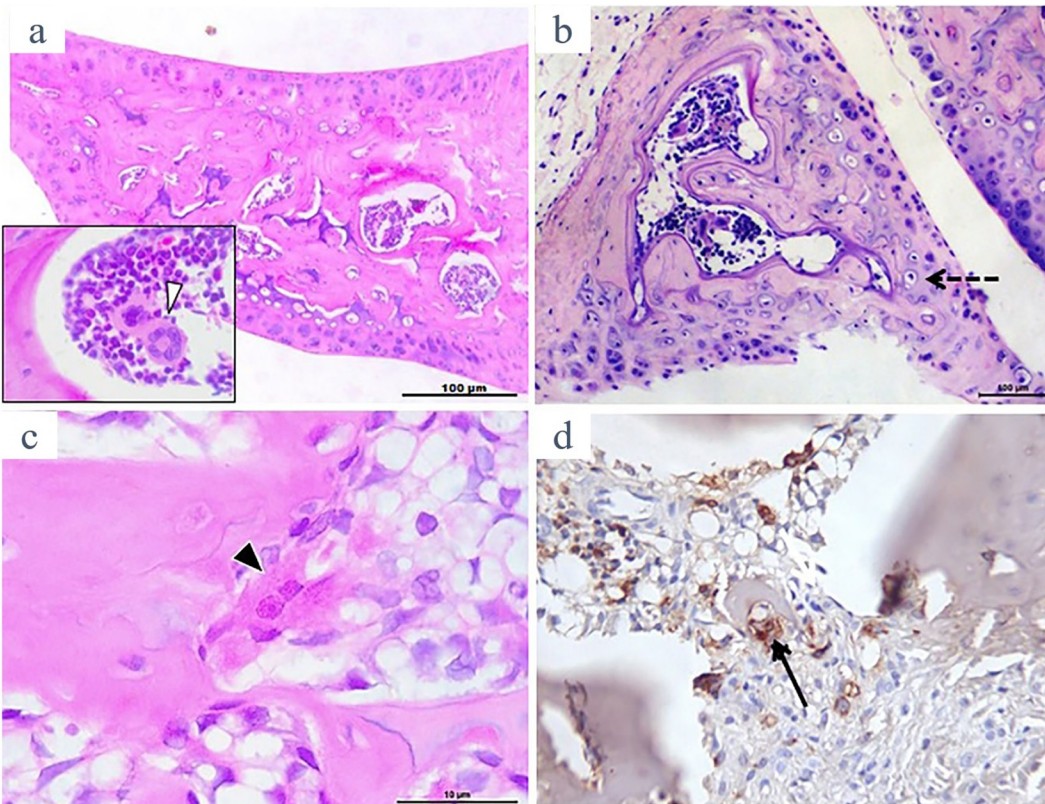

**Fig 1. Representative illustration of H&E stained rat and mouse menisci showing calcified matrix with bone marrow spaces inside (a and b, respectively; original x 100).** Inset in "a" shows a multilobed large megakaryocyte (white arrow-head; original x 1000). Inset in "b" shows a "buried" osteocyte with its surrounding lacunar space (dashed arrow). A rat meniscus sample showing an apparently active osteoclast is depicted in "c" (black arrow-head; original x 1000) and various tartrate-resistant acid phosphatase (TRAP)$^+$ stained cells indicating active, bone-resorbing multinucleated osteoclasts are shown in "d" (black arrow; original x 400).

A previous report showed that spontaneous OA developing in guinea pigs is associated with ossification inside the menisci of older animals [12], a finding that the authors attributed solely to an aging process secondary to mechanical damage. Our data question this assumption, given that the percentage of samples with ossified areas in mice increased until animals were 6 weeks-old, while decreasing in older animals of this species. However, 11 months-old rats presented the highest prevalence of ossified areas with bone marrow inside menisci. Thus, in addition to a possible species variation it is likely that aging may not be the only reason for the development of those structures.

Bone marrow cells are precursors of inflammatory cells. For instance, flushing of the marrow of long bones can be used as a method to isolate monocytes that can be later differentiated into fully mature osteoclasts capable of releasing inflammatory mediators [14]. In addition to contributing to the morphological detailing of the meniscal histology *per se*, our results have implications regarding the development of joint lesions. The meniscus, a structure that once could be regarded as an innocent bystander, is being increasingly considered to play a major role in inflammatory lesions occurring in diarthrodial joints[2,3]. It was previously shown that CD68$^+$ cells, used as a marker for macrophages, are present in the peripherally more vascularized areas of the meniscus, being absent in the central portions [7]. That study was performed using 15 human meniscus samples and apparently some sections were performed at random,

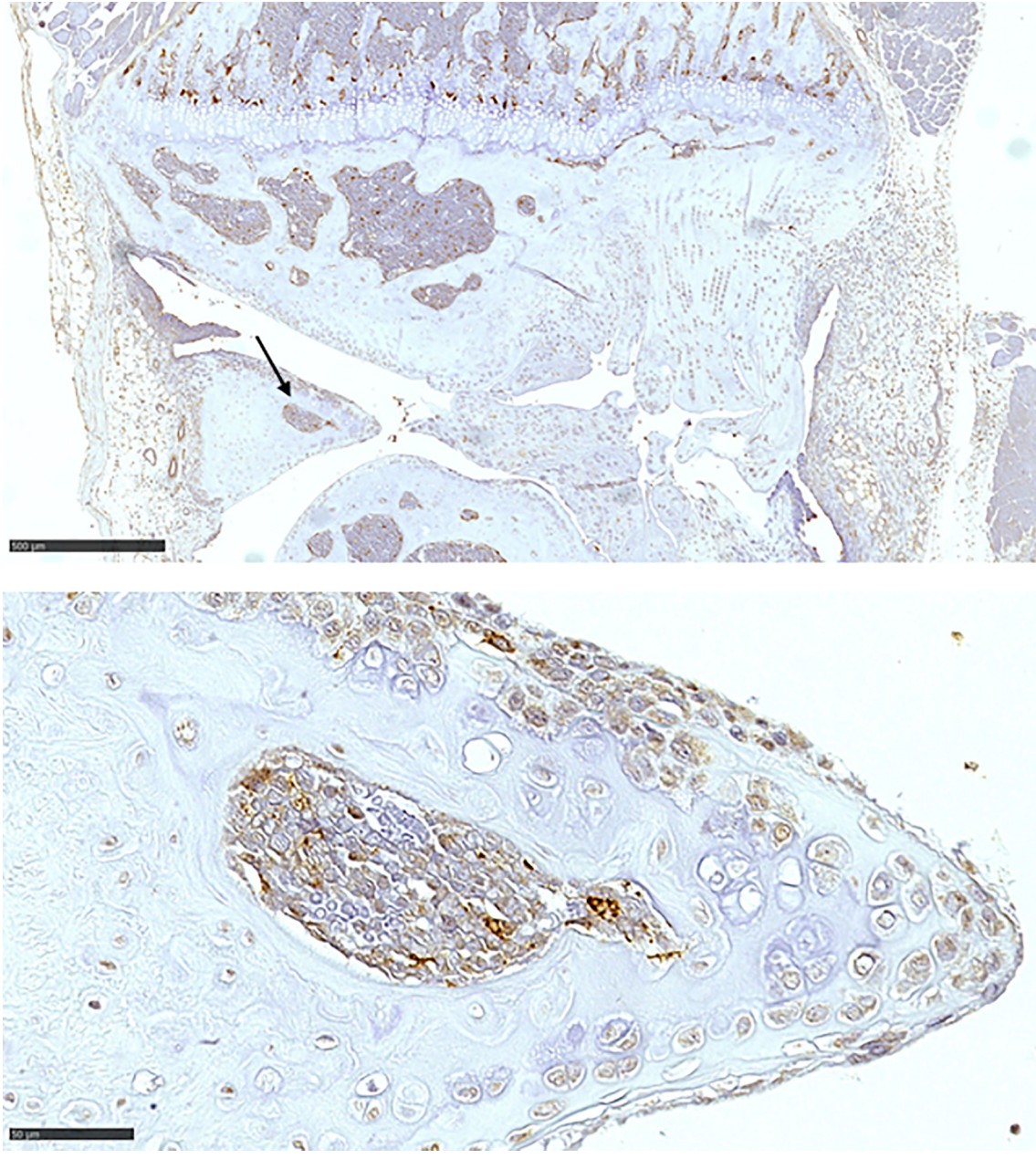

**Fig 2. Representative illustration of CD68⁺ stained cells in the meniscus of a naïve mouse.** (a) shows staining in cells within the subchondral bone as well as in bone marrow areas inside the meniscus (arrow; original x 50); (b) shows positive staining in cells in the meniscus border as well as in cells inside the bone marrow space (original x 400); bar indicates 50μM.

rather than a detailed "slicing" strategy aiming to specifically detect singularities of the meniscal cell populations. Similarly, we detected a relevant CD68⁺ staining in cells of the red zone but we also found it in cells localized inside the bone marrow areas. The occurrence of cells able to initiate and perpetuate an inflammatory reaction inside the meniscus, as those CD68⁺ macrophages, lead us to speculate that in addition to the mechanical derangement and joint instability represented by meniscal damage, cellular components of the meniscus may also trigger and/or potentiate inflammation inside joints following an insult. Chondrocyte

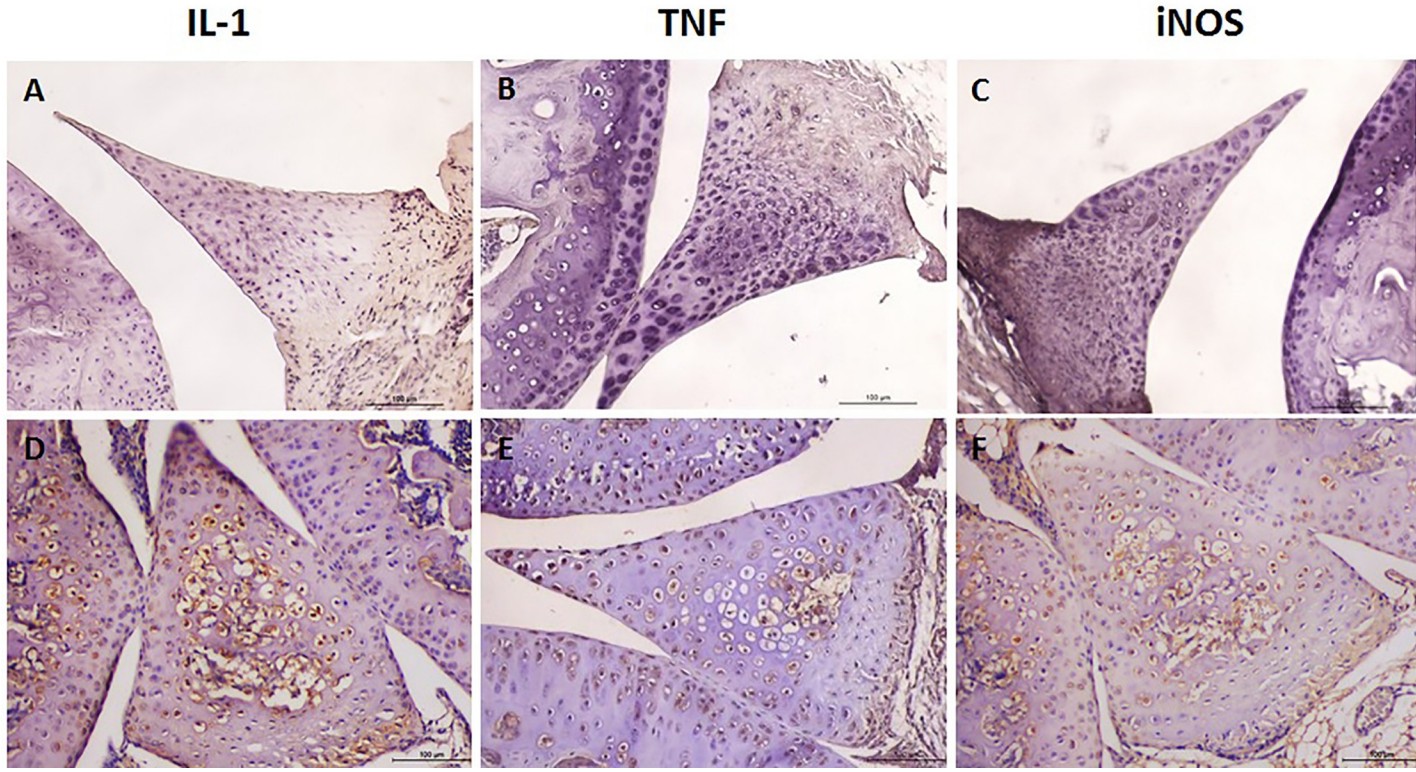

**Fig 3. Representative illustration of interleukin (IL)-1β, tumor necrosis factor (TNF) and inducible nitric oxide synthase (iNOS) immunoexpression in the meniscus of naïve mice.** While the inner parts without bone marrow areas show no staining (a,b,c), various cells inside bone marrow spaces stain positive (d,e,f); original × 400.

activation subsequent to the release of mediators produced by resident or migrated synovial cells and/or by cells entrapped in the subchondral bone [15] may be also primed by inflammatory mediators, now shown also to be possibly released from the meniscus. Accordingly, we were able to demonstrate that meniscal cells, particularly those located inside the bone marrow-like spaces within ossified areas, express inflammatory cytokines and have iNOS immunostaining. Reactive nitrogen species, including nitric oxide, have long been demonstrated to participate in OA pathophysiology [16]. Via combination with other species, such as the superoxide anion, nitric oxide may also generate peroxynitrite, which has been linked to damage to chondrocytes and osteoblasts [17,18]. To date, synoviocytes and cells harbored in the subchondral bone were regarded as the major resident sources of nitric oxide generation inside the joint [16]. Our data add meniscal cells as another possible nitric oxide provider, in this case especially in menisci from an OA animal model.

The fact that IL-1β and TNF immunostaining were demonstrated in meniscal cells does also merit consideration. Cytokine expression in naïve samples appeared to be more prominent in cells located in the bone marrow cavities, whereas both fibrochondrocytes immersed in the fibrous collagen bundles and cells in the bone marrow spaces stained positive in samples from the OA animal model, whist not occurring in the respective sham-operated group. IL-1β and TNF are major mediators in various joint disorders, not only through their direct effects, but also via inducing the release and activation of other inflammatory mediators, being crucial in the initiation and/or perpetuation of an inflammatory insult [19,20].

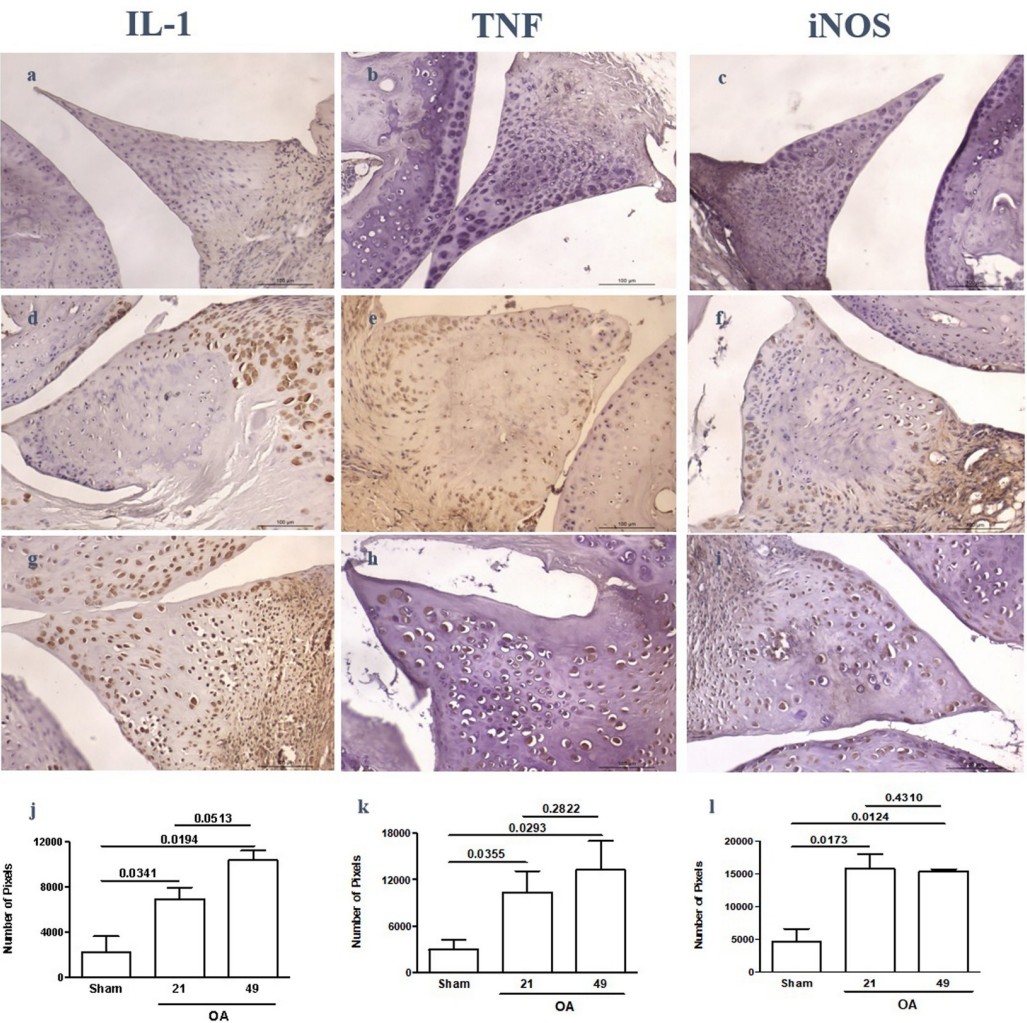

**Fig 4. Representative illustration of the immunoexpression of IL-1β, TNF, and iNOS in mouse menisci subjected to postsurgical experimental OA or a sham procedure.** While mostly cells closer to the "red zone" of the meniscus stain positive in sham samples (a,b,c) there is diffuse immunoexpression in samples from OA groups sacrificed 21 (d,e,f) or 49 (g,h,i) days after surgery. Quantitation of immunoexpression from 3 independent experiments reveals significantly increased immunoexpression in OA samples, as compared to sham (j,k,l); original magnification × 400.

Limitations of this study include the difficulty in handling samples from less than 2 months-old mice. During processing, the entire meniscus could not be entirely identified in some samples. Additionally, decalcification affected antigen retrieval when trying to better define individual cells using immunohistochemistry. The age range of the animals was based on an attempt to cover skeletal maturity in both mouse and rat species, estimated to be at 10 weeks and 3 months, respectively [21].

Given the phylogenetically conserved appearance of the knee anatomy and histology, regardless of being biped or quadruped animals [22], and the relevance of animal models to study OA pathogenesis [23], the description that inflammatory cells are present in rodent menisci, which can "inadvertently" gain access to the joint, following trauma or other events, further substantiate the relevance of the meniscus in the development of OA or in an amplification loop of inflammatory diseases such as rheumatoid arthritis. Although speculative, if those rodent changes are reproduced in humans, patients subjected to knee trauma or damage

secondary to daily life activities who present inflammatory cells inside the meniscus may be more prone to develop rapid and severe OA changes as compared to those with meniscus displaying merely fibrocartilage containing fibroblasts and fibrochondrocytes. This possibility justifies a need to search for those changes in human menisci. We are currently pursuing this objective.

## Author Contributions

**Conceptualization:** Francisco Airton Castro Rocha, João Eurico Fonseca.

**Data curation:** Francisco Airton Castro Rocha, Virgínia Claudia Carneiro Girão, Rodolfo de Melo Nunes, Ana Carolina Matias Dinelly Pinto, Bruno Vidal, João Eurico Fonseca.

**Formal analysis:** Francisco Airton Castro Rocha, Virgínia Claudia Carneiro Girão, Ana Carolina Matias Dinelly Pinto, Bruno Vidal, João Eurico Fonseca.

**Investigation:** Francisco Airton Castro Rocha, Virgínia Claudia Carneiro Girão, Ana Carolina Matias Dinelly Pinto.

**Methodology:** Francisco Airton Castro Rocha, Virgínia Claudia Carneiro Girão, Rodolfo de Melo Nunes, Ana Carolina Matias Dinelly Pinto, Bruno Vidal.

**Project administration:** Francisco Airton Castro Rocha, Virgínia Claudia Carneiro Girão.

**Supervision:** Francisco Airton Castro Rocha, João Eurico Fonseca.

**Validation:** Francisco Airton Castro Rocha.

**Visualization:** Francisco Airton Castro Rocha, Virgínia Claudia Carneiro Girão.

**Writing – original draft:** Francisco Airton Castro Rocha, Virgínia Claudia Carneiro Girão, Rodolfo de Melo Nunes, Ana Carolina Matias Dinelly Pinto, Bruno Vidal, João Eurico Fonseca.

**Writing – review & editing:** Francisco Airton Castro Rocha, Virgínia Claudia Carneiro Girão, Rodolfo de Melo Nunes, Ana Carolina Matias Dinelly Pinto, Bruno Vidal, João Eurico Fonseca.

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
