## [Decision Letter · Decision Letter 0]

4 Dec 2019

PONE-D-19-28554

Cell sources of inflammatory mediators present in bone marrow areas inside the meniscus

PLOS ONE

Dear DR. Rocha,

Thank you for submitting your manuscript to PLOS ONE. After careful consideration, we feel that it has merit but does not fully meet PLOS ONE’s publication criteria as it currently stands. Therefore, we invite you to submit a revised version of the manuscript that addresses the points raised during the review process.

We would appreciate receiving your revised manuscript by Jan 18 2020 11:59PM. To enhance the reproducibility of your results, we recommend that if applicable you deposit your laboratory protocols in protocols.io, where a protocol can be assigned its own identifier (DOI) such that it can be cited independently in the future. For instructions see: http://journals.plos.org/plosone/s/submission-guidelines#loc-laboratory-protocols

We look forward to receiving your revised manuscript.

Kind regards,

Oreste Gualillo, PharmD, PhD

Academic Editor

PLOS ONE

Journal Requirements:

2. At this time, we request that you  please report additional details in your Methods section regarding animal care, as per our editorial guidelines. Specifically,  Please describe the post-operative care received by the animals, including the frequency of monitoring for the 21 or 49 days after the surgical procedure and the criteria used to assess animal health and well-being.

In addition, please state specifcally in the Methods section of the manuscript that the mice were euthanized using an overdose of ketamine and xylazine

Thank you for your attention to these requests.

3. To comply with PLOS ONE submission guidelines, in your Methods section, please provide additional information regarding your statistical analyses. For more information on PLOS ONE's expectations for statistical reporting, please see https://journals.plos.org/plosone/s/submission-guidelines.#loc-statistical-reporting.

Additional Editor Comments:

Dear Authors,

Your paper has been reviewed by one reviewer and by myself as academic editor of the journal. There are some minor issues that should be improved in order to accept your article for publication in PLOSOne. Please, stress in the introduction the rationale of the study, as well as in the discussion.

As editor, I am particularly concerned about the clinical implications of your results. I appreciate that you will discuss more these aspects in your discussion. Your paper will be reviewed again after these minor aspects are issued.

Thank you for submit youyr results to PLOSONE

Dr. Oreste Gualillo,

PLOSONE Academic Editor.

Reviewers' comments:

Reviewer's Responses to Questions

**Comments to the Author**

1. Is the manuscript technically sound, and do the data support the conclusions?

Reviewer #1: Yes

2. Has the statistical analysis been performed appropriately and rigorously? 

Reviewer #1: I Don't Know

3. Have the authors made all data underlying the findings in their manuscript fully available?

Reviewer #1: Yes

4. Is the manuscript presented in an intelligible fashion and written in standard English?

Reviewer #1: Yes

5. Review Comments to the Author

Reviewer #1: The manuscript by Rocha et al aims to assess the cell sources of proinflammatory mediators in the meniscus of rodent animals at different ages, and to study the effect of OA in the presence of these mediators in the menisci. The paper is well designed and the experiments correctly carried out. However, some aspects need to be clarified:

* The specific objective of this manuscript is only stated in the first sentence of the abstract. It needs to be also included in the paper, preferably at the end of the introduction/background section. Furthermore, authors should emphasize, also in the sentence in the abstract, the study is only focused on rodent meniscus.

* Although macrophages have been already identified in this tissue in adult humans, the presence of this bone marrow areas inside the meniscus has been only described in animals. In my opinion, this and other apparently minor anatomical differences between rodents and humans could in some way be responsible for the lack of reproducibility of the experimental studies in comparison to the results in patients. Could authors comment on this?

* Although the age of the animals is correctly indicated in the paper, authors might include the age of skeletal maturation in each specie.

* Did authors assess whether OA increased the presence of bone marrow areas in mice, or the extension of these areas?

6. PLOS authors have the option to publish the peer review history of their article (what does this mean?). If published, this will include your full peer review and any attached files.

Reviewer #1: Yes: Raquel Largo

---

## [Author Response · Author response to Decision Letter 0]

5 Dec 2019

Answers to Editor’s and Reviewer’s comments:

PONE-D-19-28554

Cell sources of inflammatory mediators present in bone marrow areas inside the meniscus

Comment:

Journal Requirements:

Answer: Style was rechecked

Comment:

2. At this time, we request that you please report additional details in your Methods section regarding animal care, as per our editorial guidelines. Specifically, Please describe the post-operative care received by the animals, including the frequency of monitoring for the 21 or 49 days after the surgical procedure and the criteria used to assess animal health and well-being.

In addition, please state specifcally in the Methods section of the manuscript that the mice were euthanized using an overdose of ketamine and xylazine

Answers: Please see the modified methods section, as follows: Both the sham and operated groups of mice were evaluated every other day following the surgical procedure, during routine cleaning of the cages, checking for possible behavior alterations. Body weight, food and water consumption were checked weekly until the end of the protocol. Euthanasia of the animals was processed under terminal anesthesia (i.m. ketamine/xylazine overdose). Those subjected to the surgical protocol were euthanized also under terminal anesthesia (i.m. ketamine/xylazine overdose) 21 or 49 days after the surgical procedure and had their knee joints evaluated under histopathology.

Comment:

3. To comply with PLOS ONE submission guidelines, in your Methods section, please provide additional information regarding your statistical analyses. For more information on PLOS ONE's expectations for statistical reporting, please see https://journals.plos.org/plosone/s/submission-guidelines.#loc-statistical-reporting.

Answer: The name and version of the software package used for statistical analysis was included, as follows: Statistical analysis of the data was performed using SPSS Statistics for Windows, Version 16.

Additional Editor Comments:

Dear Authors,

Your paper has been reviewed by one reviewer and by myself as academic editor of the journal. There are some minor issues that should be improved in order to accept your article for publication in PLOSOne. Please, stress in the introduction the rationale of the study, as well as in the discussion. As editor, I am particularly concerned about the clinical implications of your results. I appreciate that you will discuss more these aspects in your discussion. Your paper will be reviewed again after these minor aspects are issued.

Thank you for submit youyr results to PLOSONE

Dr. Oreste Gualillo,

PLOSONE Academic Editor.

Answer: Dear Dr. Gaulillo, 

We greatly appreciate your interest and comments. Being rheumatologists, we surely are also concerned about clinical implications. This was the reason to include this sentence in the original discussion section: In addition to contributing to the morphological detailing of the meniscal histology per se, our results have implications regarding the development of joint lesions....The occurrence of cells able to initiate and perpetuate an inflammatory reaction inside the meniscus, as those CD68+ macrophages, lead us to speculate that in addition to the mechanical derangement and joint instability represented by meniscal damage, cellular components of the meniscus may also trigger and/or potentiate inflammation inside joints following an insult. 

Trying to meet your request, we added another comments in the text, as follows:

- rationale (end of introduction section): During the observation of knee samples from mice subjected to experimental OA following meniscotomy, we found defined bone marrow cavities in some surgically lesioned menisci (our unpublished data). Given the inflammatory potential of bone marrow derived cells, we decided to perform an exhaustive evaluation of menisci obtained from mice and rat knees. Our data revealed active osteoclasts...

- clinical implication (end of discussion section): Although speculative, if those rodent changes are reproduced in humans, patients subjected to knee trauma or damage secondary to daily life activities who present inflammatory cells inside their meniscus may be more prone to develop rapid and severe OA changes as compared to those with meniscus displaying merely fibrocartilage containing fibroblasts and fibrochondrocytes. This possibility justifies a need to search for those changes in human menisci. We are currently pursuing this objective. 

Comment:

Reviewer #1: The manuscript by Rocha et al aims to assess the cell sources of proinflammatory mediators in the meniscus of rodent animals at different ages, and to study the effect of OA in the presence of these mediators in the menisci. The paper is well designed and the experiments correctly carried out. However, some aspects need to be clarified:

Comment: * The specific objective of this manuscript is only stated in the first sentence of the abstract. It needs to be also included in the paper, preferably at the end of the introduction/background section. Furthermore, authors should emphasize, also in the sentence in the abstract, the study is only focused on rodent meniscus.

Answer: Thank you for this comment. We have introduced the proposed modifications, as follows:

- Abstract: Purpose: to demonstrate the production of inflammatory mediators by cells located in bone marrow spaces inside rodent menisci. 

- rationale (end of introduction section): During the observation of knee samples from mice subjected to experimental OA following meniscotomy, we found defined bone marrow cavities in some surgically lesioned menisci (our unpublished data). Given the inflammatory potential of bone marrow derived cells, we decided to perform an exhaustive evaluation of menisci obtained from mice and rat knees. Our data revealed active osteoclasts...

In order to meet an Editor’s request, we also included a speculative comment on the clinical implication. We believe this reinforces that we actually dealt with rodents, hopefully meeting your request. Please see:

- clinical implication (end of discussion section): Although speculative, if those rodent changes are reproduced in humans, patients subjected to knee trauma or damage secondary to daily life activities who present inflammatory cells inside their meniscus may be more prone to develop rapid and severe OA changes as compared to those with meniscus displaying merely fibrocartilage containing fibroblasts and fibrochondrocytes. This possibility justifies a need to search for those changes in human menisci. We are currently pursuing this objective. 

Comment:* Although macrophages have been already identified in this tissue in adult humans, the presence of this bone marrow areas inside the meniscus has been only described in animals. In my opinion, this and other apparently minor anatomical differences between rodents and humans could in some way be responsible for the lack of reproducibility of the experimental studies in comparison to the results in patients. Could authors comment on this?

Answer: Thank you for this comment. We believe there should not be great morphological (macro and micro) differences between human and rodent knees. This was the reason for our comment on the discussion, quoting prior studies: Given the phylogenetically conserved appearance of the knee anatomy and histology, regardless of being biped or quadruped animals [21], and the relevance of animal models to study OA pathogenesis..

Perhaps the main difficulty in reproducing data is due to mechanical impact (4 vs 2 stance gait), virtually no obesity in experimental animals, or other less obvious reasons. But this remains to be proven. There are some issues to get human data. Usually, human samples are just thrown away following meniscectomy in elderly patients; during arthroscopy, they are minced to tiny pieces. On the other hand, in the young patient, following trauma, surgeons preserve as much meniscus as possible. During arthroplasty, there is almost no meniscus left. The best chance would be to perform an active search using cadavers. We are pursuing this strategy.

Comment:* Although the age of the animals is correctly indicated in the paper, authors might include the age of skeletal maturation in each specie.

Answer: Thank you for this comment. Actually, our ethics committee limited the number of animals and we explained our intention to cover periods pre and post skeletal maturation. Please see the discussion section (another ref was included), as follows: The age range of the animals was based on an attempt to cover skeletal maturity in both mouse and rat species, estimated to be at 10 weeks and 3 months, respectively [21]

Comment:* Did authors assess whether OA increased the presence of bone marrow areas in mice, or the extension of these areas?

Answer: Thank you for this very relevant comment. We really wanted to do that. With the number of animals we got that was impossible. We can’t know in advance (prior to the start of experiments) which animals will present those areas. This means we would need more samples. We had asked to include 15 animals per OA group, based on a sample calculation (however, merely estimating) but our committee on animal experimentation did not allow that. 

6. PLOS authors have the option to publish the peer review history of their article (what does this mean?). If published, this will include your full peer review and any attached files.

Do you want your identity to be public for this peer review? For information about this choice, including consent withdrawal, please see our Privacy Policy.

Reviewer #1: Yes: Raquel Largo

---

## [Editor Report · Decision Letter 1]

11 Dec 2019

Cell sources of inflammatory mediators present in bone marrow areas inside the meniscus

PONE-D-19-28554R1

Dear Dr. Rocha,

We are pleased to inform you that your manuscript has been judged scientifically suitable for publication and will be formally accepted for publication once it complies with all outstanding technical requirements.

With kind regards,

Oreste Gualillo, PharmD, PhD

Academic Editor

PLOS ONE

Additional Editor Comments (optional):

No further comments
---

## [Editor Report · Acceptance letter]

12 Dec 2019

PONE-D-19-28554R1 

Cell sources of inflammatory mediators present in bone marrow areas inside the meniscus 

Dear Dr. Rocha:

I am pleased to inform you that your manuscript has been deemed suitable for publication in PLOS ONE. Congratulations! Your manuscript is now with our production department. 

With kind regards,

on behalf of

Dr Oreste Gualillo 

Academic Editor

PLOS ONE